# Small RNA and Transcriptome Sequencing Reveals miRNA Regulation of Floral Thermogenesis in *Nelumbo nucifera*

**DOI:** 10.3390/ijms21093324

**Published:** 2020-05-08

**Authors:** Yu Zou, Guanglong Chen, Jing Jin, Ying Wang, Meiling Xu, Jing Peng, Yi Ding

**Affiliations:** 1State Key Laboratory of Hybrid Rice, Department of Genetics, College of Life Sciences, Wuhan University, Wuhan 430072, China; zouxiaoyu@whu.edu.cn (Y.Z.); glchen@whu.edu.cn (G.C.); jinjing1130@whu.edu.cn (J.J.); yingwang@whu.edu.cn (Y.W.); meilingxu@whu.edu.cn (M.X.); 2Institute of Vegetable, Wuhan Academy of Agricultural Science, Wuhan 430065, China; pengjing67@163.com

**Keywords:** *Nelumbo nucifera*, floral thermogenesis, receptacle, miRNA sequencing, transcriptome sequencing

## Abstract

The sacred lotus (*Nelumbo nucifera* Gaertn.) can produce heat autonomously and maintain a relatively stable floral chamber temperature for several days when blooming. Floral thermogenesis is critical for flower organ development and reproductive success. However, the regulatory role of microRNA (miRNA) underlying floral thermogenesis in *N. nucifera* remains unclear. To comprehensively understand the miRNA regulatory mechanism of thermogenesis, we performed small RNA sequencing and transcriptome sequencing on receptacles from five different developmental stages. In the present study, a total of 172 known miRNAs belonging to 39 miRNA families and 126 novel miRNAs were identified. Twenty-nine thermogenesis-related miRNAs and 3024 thermogenesis-related mRNAs were screened based on their expression patterns. Of those, seventeen differentially expressed miRNAs (DEMs) and 1765 differentially expressed genes (DEGs) had higher expression during thermogenic stages. The upregulated genes in the thermogenic stages were mainly associated with mitochondrial function, oxidoreductase activity, and the energy metabolism process. Further analysis showed that miR156_2, miR395a_5, miR481d, and miR319p may play an important role in heat-producing activity by regulating cellular respiration-related genes. This study provides comprehensive miRNA and mRNA expression profile of receptacle during thermogenesis in *N. nucifera*, which advances our understanding on the regulation of floral thermogenesis mediated by miRNA.

## 1. Introduction

Several plant taxa can produce heat autonomously to ensure that the flower chamber temperature is higher than ambient temperatures. *Nelumbo nucifera* [1], *Philodendron selloum* [2], *Dracunculus vulgaris* [3], and *Magnolia ovata* [4] are reported to be thermogenic plants. Extensive research indicates that thermogenesis has critical biological significance. The suitable temperature of the chamber provides a warm environment for flower organ development and facilitates fertilization [5,6]. Volatile floral scent compounds under high temperature can also attract insect pollinators to ensure fertilization success [7].

Although the phenomenon of floral thermogenesis was discovered many years ago, the regulatory molecular mechanism of floral thermogenesis remains unclear. Previous research suggested that heat generation was associated with cyanide-resistant respiration mediated by an alternative oxidase (AOX) [7]. Alternative pathway flux, measured by oxygen isotope discrimination techniques, significantly increased in the *N. nucifera* receptacle during the thermogenic stage [8]. AOX is a terminal oxidase of the mitochondrial respiratory chain. AOX can couple and oxidize ubiquinol, which allows electrons to bypass complexes III and IV of the cytochrome pathway and to be directly delivered to oxygen [9]. Because some proton-pumping steps are avoided in the cyanide-resistant respiration pathway, the synthesis of ATP is also reduced, and massive amounts of energy are released as heat [10]. In addition to AOX, a plant uncoupling protein (UCP) located in the mitochondrial inner membrane is another energy dissipating system. UCP can transport protons from the membrane gap to the mitochondrial matrix, which bypasses ATP synthesis. Therefore, chemical energy is converted into heat [11,12]. The key regulators of thermogenesis are different between species [13]. Previous studies showed that the specific expression pattern of *AOX* and *UCP* seemed to depend on respiratory substrates. If the respiratory metabolic substrate was carbohydrate, the *AOX* gene was expressed; if the substrate was lipid, the *UCP* gene was expressed [14]. As these two energy-dissipation-related genes are also expressed in nonthermogenic plants [15], there may be additional genes associated with floral thermogenesis. To determine the molecular basis of thermogenesis more comprehensively, transcriptome analysis was performed in several thermogenic plants [13,16]. In skunk cabbage, the gene expression profiles of spadix were examined, and the genes related to maintenance and termination of thermogenesis were identified [16]. In *Arum concinnatum*, a total of 1266 transcripts were screened, and the expression patterns of these transcripts were significantly correlated with temperature trends for each sample [13]. In the process of thermogenesis, the expression of some genes changed rapidly, which may be involved in epigenetic regulation [17].

MicroRNA (miRNA), a noncoding RNA, plays an important role in the post-transcriptional regulation of genes, mainly by mediating gene silencing [18]. Many studies have shown that miRNAs are involved in the flowering related process [19]. In cotton, ghr-miR393, At_chr9_3080, and Dt_chr12_6065 take part in anther development by regulating the target genes involved in the cell cycle, auxin anabolic processes, and carbohydrate metabolism [20]. In Arabidopsis, miR319a can affect petal growth by targeting *TCP4* [21]. In wheat, tae-miR1127a and tae-miR2275 may be associated with wheat male sterility by regulating the *SMARCA3L3* and *CAF1* genes, respectively [22]. Because autonomous heat production is a unique phenomenon to some flowering plants, and the process of thermogenesis is also coupled to the floral development process, research on whether miRNAs participate in floral thermogenesis has been carried out. Seventeen differentially expressed miRNAs were identified in *M. denudate* pistils between the nonthermogenic and the thermogenic stages, of which the target genes were mainly enriched in ‘polyprenyl transferase activity’ and ‘photosynthetic electron transport’. Therefore, these miRNAs were thought to be involved in thermogenesis of *M. denudate* by regulating cellular respiration and light reactions [17]. However, the role of miRNA in the floral thermogenesis of *N. nucifera* has not yet been reported.

*N. nucifera* is an ancient thermoregulatory dicotyledon that maintains its chamber temperature between 32 °C and 35 °C for several days during flowering in spite of changing environmental temperatures [23]. The development of *N. nucifera* flowers can be divided into five periods: from stage 1 to stage 5 [24,25]. In stage 1, the flower is just a small green bud in a prethermogenic state. In stage 2, the flower bud grows, and the petal tips become pink. A significant amount of heat starts to be produced at this stage. In stage 3, the petals open 2–12 cm, and the protruding stigmas of the flower are receptive, while the immature stamens are tightly attached to the receptacle; the heat production reaches a peak at this stage. In stage 4, the petals are horizontal, and numerous stamens bearing mature pollen are released. Then, the petals and stamens abscise, and thermogenesis cannot be detected in stage 5. Three distinct physiological phases can be identified during flower development of *N. nucifera*: prethermogenic (stage 1), thermogenic (stages 2–4), and post-thermogenic (stage 5) [24,25]. In *N. nucifera*, the receptacle was found to have strong thermoregulatory activity, which provided suitable material to investigate the molecular mechanism of thermogenesis [26]. Moreover, AOX protein content and alternative pathway flux were synchronized with thermogenic activity during receptacle development. The cytochrome oxidase pathway contributed little to increased respiratory flux. Therefore, AOX rather than UCP plays a role in the thermogenesis of *N. nucifera* [25]. The regulation of AOX activity in the thermogenic receptacle of *N. nucifera* has also been investigated. Both AOX isoforms from *N. nucifera* lack Cys_1_. Reduced Cys_1_ can take part in the post-translational regulation of AOX by interacting with a-keto acids such as pyruvate. Therefore, the activity of AOX was stimulated by succinate rather than pyruvate or glyoxylate [27]. In *N. nucifera*, the warm receptacle temperature can provide heat rewards for insects, thereby promoting pollen transmission and pollination [23]. Moreover, since the pistil is wrapped in the receptacle, a stable receptacle temperature is conducive to fertilization success and seed development. For example, previous studies have shown that lower temperatures can affect postpollination events and decrease seed production in lotus flower [5]. Overall, *N. nucifera* has evolved a floral thermogenesis strategy to maximize its reproductive success, and the investigation of thermogenesis has critical biological significance.

In the present study, to comprehensively understand the molecular basis and miRNA regulatory mechanism of thermogenesis in *N. nucifera*, a transcriptome sequencing library and a small RNA sequencing library from receptacle tissue at stages 1, 2, 3, 4, and 5 were constructed, and the miRNA expression profiles and gene expression profiles were obtained during the floral thermogenesis process. In addition, the putative thermogenesis-related miRNA and mRNA were identified based on the correlation between the expression level and temperature fluctuation. Small RNA sequencing combined with the transcriptome sequencing in *N. nucifera* in this study lays a solid foundation for the in-depth study of floral thermogenesis mechanisms in the future.

## 2. Results

### 2.1. Small RNA Sequences in the Receptacle of N. nucifera

To visualize the phenomenon of floral thermogenesis in *N. nucifera*, flowers in the thermogenic stage (stage 4) grown in the natural environment were photographed with an infrared camera system FLIR-SC660 (FLIR SYSTEMS, USA) (Figure 1). The infrared image showed that the temperature of the receptacle was significantly higher than the ambient temperature, indicating that the receptacle was the heat-producing region, which is consistent with previous studies [28]. Detailed measurement results of receptacle temperature and ambient temperature were obtained (Appendix A). Therefore, the receptacle was a suitable material to investigate floral thermogenesis in *N. nucifera*. To obtain a comprehensive miRNA expression profile for the *N. nucifera* receptacle, a total of 15 small RNA-seq libraries were constructed from receptacles of the five chronological development stages, each stage with three biological replicates. The average raw reads obtained from stage 1 to stage 5 were 29,013,878, 27,523,439, 29,550,480, 29,902,369, and 28,617,333, respectively, in our RNA-seq data. After removing low-quality reads, the average clean reads obtained at each stage were 27,301,352, 25,844,417, 27,318,365, 23,909,875, and 24,986,799, respectively. The average ratio between the clean tags and the raw tags was 89.52% for all samples. Approximately 85.22% clean tags were successfully aligned to the *N. nucifera* reference genome (Appendix A). Meanwhile, 172 known miRNAs (belonging to 39 miRNA families) were detected by mapping clean tags to the miRbase database (Appendix A), and 126 novel miRNAs were predicted based on the architectural features of those unannotated sRNA tags (Appendix A). Among the 172 known miRNAs, 140 were expressed along the whole anthesis without stage specificity (Figure 2A). One, two, and four known miRNAs were expressed specifically at stages 1, 3, and 5, respectively. Only one miRNA, miR393a_3, was shared by stages 2, 3, and 4. We found that stages 2 and 4 have no unique expression of known miRNAs (Figure 2A and Appendix A). Then, the differential expression miRNAs (DEMs) in different stages were identified based on the screening criteria of number fold change ≥ 2 and Q-value ≤ 0.001. In total, 138 DEMs between any two diverse stages were screened out (Figure 2B).

### 2.2. Time-Series miRNA Expression Analysis

Because the sequencing materials come from five continuous development stages, time-series expression analysis was carried out to reveal the temporal expression pattern of miRNA in the study by using the Mfuzz package [29]. All miRNAs were classified into nine clusters (Figure 3) and their corresponding members are listed in Appendix A. Among the nine clusters, clusters 3, 4, and 9 showed a similar expression trend where miRNA expression levels increased with the development of the *N. nucifera* flower. Meanwhile, cluster 2 had a completely opposite trend of gradually decreasing expression during this process. The other five clusters had no consistent trend with the flower development process. However, the average expression level of miRNAs at each stage in clusters 7 and 8 formed a clear tendency related to heat production trends during thermogenesis in *N. nucifera*. In cluster 7, the miRNAs had a higher average expression level in the thermogenic stage than in the nonthermogenic stage (Figure 4A), while the miRNAs had a lower expression level during the thermogenic stage in cluster 8 (Figure 4B). Therefore, it can be inferred that the miRNAs classified into these two clusters may play a role in the thermogenesis of *N. nucifera*.

Go enrichment analysis and KEGG analysis of the target genes in clusters 7 and 8 were performed to understand their biological function. Cluster 7 contains 350 target genes of 17 DEMs (Appendix A). The top five enriched molecular function terms were nucleic acid binding, DNA binding, heterocyclic compound binding, organic cyclic compound binding, and sulfate adenylyltransferase (ATP) activity (Appendix A). KEGG enrichment results showed that ubiquinone and other terpenoid−quinone biosynthesis, riboflavin metabolism, and pentose and glucuronate interconversions were significantly enriched pathways (Appendix A). In cluster 8, 352 target genes of 12 DEMs were predicted, mainly involved in membrane transport activities, such as amino acid, organic acid, carboxylic acid, and organic anion transmembrane transporter activity (Appendix A). Oxidative phosphorylation, pentose and glucuronate interconversions, and metabolism were enriched pathways (Appendix A).

### 2.3. Transcriptome Sequencing and DEG Analysis during Receptacle Development

The materials obtained from the five different development stages of the receptacle were also used for the transcriptome sequencing, and fifteen RNA libraries were constructed. The average yield of each sample was 12.60 Gb. The clean reads obtained in these five periods on average were 126,483,630, 126,108,901, 126,532,292, 126,578,510, and 124,138,839, respectively (Appendix A). A total of 59,712 expressed transcripts were detected, of which 16,178 were novel mRNAs and 20,841 were known mRNAs. After normalizing the expression of genes, differential expression genes (DEGs) can be screened by comparing FPKM between different samples. When using stage 1 as a control, the number of significant differentially expressed known genes in stages 2, 3, 4, and 5 were 2960, 5662, 5663, and 7505, respectively. When stages 3, 4, and 5 were compared with stage 2, 4810, 5490, and 7397 DEGs were screened, respectively. When stages 4 and 5 were compared with stage 3, 4811 and 6812 DEGs were identified individually. In addition, 5244 DEGs were obtained by comparing stages 4 with stage 5 (Appendix A). Moreover, when stage 3 was used as a control, the number of DEGs between the distinct stages of thermogenic period was less than that between the thermogenic stage and the nonthermogenic stage.

### 2.4. Time-Series DEG Expression Analysis

To understand the temporal and spatial expression pattern of DEGs, a time-series expression analysis was performed in this study. Similar to miRNA analysis, all DEGs were classified into nine clusters. The nine clusters and their corresponding members are listed in Appendix A. Genes in one cluster have similar expression patterns and may be involved in the same biological processes. Each cluster has a unique expression peak at the development stage five (Figure 5). We speculated that the genes whose expression trend is related to heat production tendency may participate in the floral thermogenesis process. As shown in Figure 5, except for clusters 5 and 9, the other seven clusters had no special expression bias between the thermogenic stage and the nonthermogenic stage. Our study showed that, in cluster 5, with the development of the receptacle, the gene expression levels gradually increased and reached a peak in stage 3 prior to slowly decreasing until stage 5. Overall, the genes in this cluster had higher expression in the thermogenic stage than in the nonthermogenic stage. Heat production of *N. nucifera* also follows this tendency. A total of 1765 DEGs were classified into cluster 5 (Appendix A). The AOX encoding gene (NNU_06050), which has been confirmed to play a role in floral thermogenesis in *N. nucifera* [25], was also identified in this cluster. A portion of the gene encoding the mitochondrial respiration chain complex is also contained in this cluster. For example, NADH dehydrogenase genes, Ubiquinol-cytochrome c reductase complex genes, cytochrome oxidase genes, and ATP synthase genes were included in this cluster. Different from cluster 5, the genes in cluster 9 had opposite expression trends when compared with the heat-producing pattern (Figure 5). The lowest expression showed in cluster 9, and correspondingly the highest expression in cluster 5 appeared in stage 3. A total of 1259 DEGs were assigned to this category, and some of these DEGs were related to photosynthesis (Appendix A). For example, the photosystem Ⅰ complex, the photosystem Ⅱ complex, and some other thylakoid membrane protein-encoding genes were included in this class. GO analysis and KEGG enrichment analysis were performed to further understand the biological function of DEGs in clusters 5 and 9 (Appendix A). In cluster 5, the top five enriched GO terms in the cell component category were mitochondrion, organelle membrane, mitochondrial membrane, respiratory chain, and mitochondrial envelope (Appendix A). The results of KEGG enrichment analysis also showed that the significantly enriched pathway contained cellular respiration-related pathways, such as glycolysis, pentose phosphate pathway, and tricarboxylic acid cycle (Appendix A). These results indicated that cellular respiration and mitochondrial function are closely related to floral thermogenesis. In cluster 9, the top five enriched GO terms in the cell component category were photosystem, photosystem I, photosynthetic membrane, thylakoid, and cell (Appendix A). The photosynthesis pathway was also identified at both the enriched KEGG pathway and the enriched GO terms (Appendix A). These results indicated that the receptacle is mainly used as a photosynthetic organ after terminating heat generation, which may be related to the green color of the receptacle at stage 5.

### 2.5. Quantitative RT-PCR Analysis of the miRNAs and Their Target Genes

To confirm the reliability of sequencing data, qRT-PCR was performed to analyze the expression patterns of nine randomly selected DEMs and three of their target genes. The heatmap of Figure 6 presents the results of the sequencing data, and the histogram presents the experimental results of qRT-PCR. These results showed that the expression patterns of seven miRNAs (miR845d, miR397a_3, miR399b, miR319p, miR169h_2, miR160a-5p, and miR319a-5p_1) at five continuous developmental stages were basically consistent with the high-throughput sequencing data in the study (Figure 6). For instance, the expression of miR399b reached a peak in stage 3, decreased in stage 4, and finally increased again in stage 5 (Figure 6A). miR397a_3 had the lowest expression in stage 3 and the highest expression in stage 5, which showed a general expression tendency to decrease before increasing throughout the whole development process (Figure 6B). Although the expression trends of the other two miRNAs (miR319c-2, miR166h-3p) were similar to the sequencing data, the highest expression stage was shifted. For example, the results of qRT-PCR showed that miR319c_2 had the highest expression level in stage 3, while the sequencing data showed that the highest expression period was stage 5 (Figure 6B). This offset may be caused by the algorithmic difference between qRT-PCR and sequencing. Meanwhile, the expression of three target genes (NNU_02813, NNU_03138, and NNU_09622) was also confirmed using qRT-PCR. NNU_02813, NNU_03138, and NNU_09622 are the predicted targets of miR319p, miR319a-5p_1, and miR160a-5p, respectively. As a result, the expression of these target genes was negatively correlated with the corresponding miRNAs (Figure 6C,D). The qRT-PCR results of these miRNAs are basically consistent with high-throughput sequencing data, indicating the reliability of sequencing data. Moreover, different expression patterns of these miRNAs were identified with RT-qPCR at five developmental stages of receptacle. The expression levels of four miRNAs, including miR169h_2, miR319c_2, miR160a_5p, and miR319a-5p_1, reached the peak at stage 3. The other two miRNAs, which were miR166h_3p and miR397a_3, had the highest expression in stage 5. miR399b and miR319p had similar expression trends, both of which were highly expressed in stage 3 and stage 5. There was a unique expression pattern for miR845d, which was highly expressed in stage 1 and stage 2. These results imply that different miRNAs may regulate the development of the receptacle at different stages. Therefore, miRNAs that are highly expressed during the thermogenic stages may be involved in the regulation of floral thermogenesis in *N. nucifera*.

### 2.6. Integration Analysis of the miRNAs and Genes Associated with Thermogenesis in N. nucifera

Based on the time-series analysis of miRNA, a total of 29 DEMs were identified between the thermogenic and nonthermogenic stages. By integrating the transcriptome sequencing data with the small RNA sequencing data, the miRNA–mRNA pairs whose expression was negatively related in the same difference group were screened. A total of 172 miRNA–target pairs of 23 thermogenesis-related miRNAs were identified (Appendix A) in our study. Our transcriptome data suggested that cellular respiration may play an important role in floral thermogenesis. Therefore, the miRNA–target pairs associated with these functions from these 23 miRNAs were screened. Finally, nine miRNA–target pairs from four miRNAs were obtained. miR156_2 had two targets (NNU_23701, NNU_03737) involved in glycolysis, with one target (NNU_23230) participating in the TCA cycle, and two targets (NNU_10619, NNU_14787) involved in ATP synthesis coupled with proton transport. miR395a-5 had one target (NNU_18627) participating in glycolysis and another target (NNU_26055) involved in ATPase activity. miR481d had one target (NNU_19724) associated with glycolysis, and miR319p had one target (NNU_10749) related to electron carrier activity (Figure 7). These four miRNAs may be involved in floral thermogenesis by regulating cellular respiration processes.

## 3. Discussion

The phenomenon of thermogenesis is essential for some flowing plants. It is beneficial for plants to resist cold stress and to provide a suitable temperature for the development of reproductive organs to ensure reproductive success. Comprehensive gene expression research on thermogenic plants has only been carried out in a few species, such as *A. concinnatum*, *Magnolia denudate*, and skunk cabbage. Rapidly changing gene expression patterns suggest that miRNA-mediated epigenetic regulation may be involved in this process [13,16,17]. However, little is known about the miRNAs that play a role in floral thermogenesis of *N. nucifera*, so it is of great significance to study the mechanism of floral thermogenesis in *N. nucifera* to understand the plant’s temperature responsive mechanism. *N. nucifera* is a perennial aquatic plant, which is very important in horticulture, medicinal usage, and plant phylogeny. Floral thermogenesis is unique characteristic of *N. nucifera* compared with other aquatic plants [30]. The pistil-embedded receptacle is the main region of thermogenesis [26]. Stigma receptivity of flowers coincides with the peak of thermoregulation [31]. Stable floral temperature facilitate the development of pollen and ovules, and promote fertilization and seed development. Lower temperature reduces seed production in *N. nucifera* [5]. Functional transformation of the receptacle occurs after fertilization, from a thermogenic structure at anthesis to a photosynthetic structure postanthesis, which may involve extremely complex regulatory mechanisms [32]. Therefore, transcriptome and small RNA sequencing of the receptacle are beneficial for understanding the regulatory mechanism of floral thermogenesis and functional transformation of the receptacle.

### 3.1. MiRNA-Mediated Epigenetic Regulation in the Process of Floral Thermogenesis in N. nucifera

In the present study, a total of 172 known miRNAs from 39 miRNA families were identified from the receptacle during flower development (Appendix A). The largest miRNA family identified was miR396, with 15 family members, followed by the miR171 and miR166 families, with 14 and 13 members, respectively. These miRNA families identified from the ancient dicotyledon lotus (*N. nucifera*) indicated that some miRNAs are evolutionarily conserved, which is consistent with previous studies [33]. In addition, five known miRNA families that had been reported as involved in floral organ development were also identified in *N. nucifera* during the flower development process, including miR160, miR319, miR167, miR159, and miR166 [19,34]. Of these known miRNAs, 140 miRNAs were detected in five continuous development periods, and the remaining 32 miRNAs were only expressed at certain periods. For example, miR393a_3 was only expressed in three periods: stages 2, 3, and 4, which are known as the thermogenic stages (Figure 2A). The results indicated that some miRNAs have the characteristics of spatiotemporal specific expression.

By using a time-series cluster, the miRNA expression pattern analysis was conducted, and all identified miRNAs in this research were clustered into nine categories based on their expression patterns (Figure 3). It was notable that the miRNAs in cluster 7 had higher average expression levels during the thermogenic phase (from stage 2 to stage 4), while the miRNAs in cluster 8 had lower average expression levels in the thermogenic stage (stage 1 and stage 5). The expression patterns of miRNAs in the other seven clusters had no special bias in the thermogenic stages. Therefore, the miRNAs in clusters 7 and 8 were differentially expressed between the thermogenic phase and the nonthermogenic phase. GO enrichment analysis and KEGG analysis showed that the target genes of miRNAs in cluster 7 were involved in ubiquinone biosynthesis and riboflavin metabolism (Appendix A). Both are hydrogen donors for redox reactions. Moreover, ubiquinone is an important component of the respiratory chain. Several enriched GO terms and pathways of the miRNA targets in cluster 8 were transmembrane transport, oxidative phosphorylation and some metabolic processes (Appendix A). These results suggested that miRNAs may participate in the floral thermogenesis process by regulating genes involved in the cellular respiratory process and some energy metabolism pathways.

### 3.2. Regulation of Floral Thermogenesis Has a Characteristic Feature in Time-Series and Spatiotemporal Expression Patterns

A total of 56,354 DEGs were identified between any two periods of lotus flower development in this study. When compared to stage 3 (thermogenic stage), 5662 and 6812 DEGs were detected in stage 1 (prethermogenic stage) and stage 5 (post-thermogenic stage), respectively. However, the DEGs among thermogenic stages (from stage 2 to stage 4) were as follows: when using stage 3 as a control, 4810 and 4811 DEGs were obtained in stages 2 and 4, respectively. In addition, 5490 DEGs were screened from the comparison between stages 2 and 4 (Appendix A). The number of DEGs among the thermogenic stages was less than that between the thermogenic stages and the nonthermogenic stage. Previous studies have mentioned that the regulation of AOX protein during floral thermogenesis is not constant throughout the developmental sequence and the thermogenic stages in *N. nucifera*. The coarse regulation of heat production activity throughout the developmental process was achieved by changing the protein content. Precise regulation of the thermogenic stages occurred via the post-translational level of the AOX protein. Therefore, the protein content of AOX was higher in the thermogenic stages than in the nonthermogenic stages, and the AOX protein level did not change in the three thermogenic stages [25]. The numerous DEGs identified in this study indicated that other genes in addition to *AOX* might regulate the heat production activity of *N. nucifera* in this way.

We have analyzed all DEGs based on a time-series cluster. The results showed that all DEGs were divided into nine categories (Figure 5). The gene expression pattern in clusters 5 and 9 correlated with heat production trends. A total of 3024 DEGs were screened as thermogenesis-related mRNAs, of which 1765 DEGs in cluster 5 had higher expression in the thermogenic stages, and 1259 DEGs in cluster 9 had higher expression in the nonthermogenic stages (Appendix A). In cluster 5, the *AOX* gene, which has been proven to play a role in floral thermogenesis in *N. nucifera*, was included. In addition, a large number of genes involved in the four processes of cellular respiration, the glycolytic pathway, mitochondrial respiratory chain, TCA cycle, and pentose phosphate pathway, were all identified in this cluster. GO enrichment analysis was performed to comprehensively understand the function of genes in cluster 5 (Appendix A). These results indicated that the mitochondria and the respiratory chain complex were the major enriched cellular components terms for these genes. The main enriched molecular function terms were cofactor binding, metal cluster binding, and iron–sulfur cluster binding. It is notable that many carbohydrate metabolism processes were included in the enriched biological pathway terms, such as the hexose catabolic process, monosaccharide catabolic process, and glucose catabolic process (Appendix A). Previous studies have mentioned that the respiratory substrate of *N. nucifera* seems to be carbohydrate; therefore, the content of soluble carbohydrates and starch was measured during the five development stages of the receptacle. The average content of starch was higher during the thermogenic stages than the post-thermogenic stage, which indicated that thermogenesis activities were fueled by starch [24,25]. Higher expression of these genes in the thermogenic stages may increase the cellular respiration, which leads to increased oxygen consumption and heat production to facilitate the spatiotemporal expression of developmental processes. The mechanism of floral thermogenesis in *N. nucifera* is similar to that of thermogenic skunk cabbage, which is also related to mitochondrial function and cellular respiration pathways [16]. The reason may be that *N. nucifera* and skunk cabbage are both thermoregulatory plants.

In cluster 9, the genes had a higher expression level in the nonthermogenic stages. GO enrichment analysis showed that the major enrichment terms of these genes in biological processes included photosynthesis and maintenance of floral meristem identity. The main enrichment terms in molecular function included UDP-galactose: N-glycan beta-1,3-galactosyltransferase activity, and peptide binding. It was notable that several significant enrichment terms of cellular components were photosynthetic membrane, photosystem, and thylakoid, which are closely related to photosynthesis (Appendix A). Moreover, some photosynthesis activities during the development of the receptacle have been reported previously, such as Rubisco content, photosynthetic rate, and photoprotective and photosynthetic pigments [32]. These changes in photosynthetic characteristics are related to the functional transformation of receptacle, from the thermogenesis structure at anthesis to the photosynthesis structure postanthesis [32]. Therefore, a large number of expressed genes in the nonthermogenic stage may lead to increased photosynthesis activities, thereby promoting the function transformation of the receptacle of *N. nucifera*.

### 3.3. Temperature-Responsive miRNAs Involved in the Regulation of Floral Thermogenesis in N. nucifera

By integrating the miRNA expression files with mRNA expression files, 172 miRNA–target pairs from 29 thermogenesis-related miRNAs were obtained (Appendix A). Among these 29 miRNAs, 11 known miRNA families were involved, including miR156, miR160, miR164, miR166, miR167, miR169, miR171, miR319, miR395, miR399, and miR481. Interestingly, in addition to miR481, the other 10 miRNA families were temperature-responsive miRNAs that had been reported previously. These temperature-responsive miRNAs have been investigated in many species [35,36,37]. In Arabidopsis, six environmental temperature-responsive miRNA families were identified using microarray and the northern hybridization experiments at 16 °C and 23 °C. Two miRNA families (miR156 and miR169) were upregulated at low temperatures, and the other four miRNA families (miR163, miR172, miR398, miR399) were upregulated at higher temperatures [38]. The miR399 overexpressing plant and PHOSPHATE 2 (target of miR399) mutant plants had an earlier flowering time than the wild type at 23 °C, while the flowering time was consistent with the wild type at 16 °C. The results indicated that miR399 was involved in regulating flowering time in Arabidopsis under changed external temperatures [39]. Transcriptome analysis indicated that miR159/319, miR164, miR165/166, and miR169 were cold-induced miRNAs in *Arabidopsis thaliana* [40,41]. miR160 and miR167 were identified as temperature-responsive miRNAs in cotton. The miR160 family had lower expression levels at both 12 °C and 42 °C when compared with the control conditions at 25 °C. The ARF (auxin response factor) gene was identified as the target gene of miR160. Therefore, miR160 is involved in regulating seedling growth under temperature pressure [42]. In soybeans, qRT-PCR experiments confirmed a significant increase in the expression of gma-miR166u and gma-miR171p, gma-miR167c, and gma-miR399 under cold treatment conditions [43]. In *Dactylis glomerata*, the expression of miR395 was significantly higher in the vernalization stage than before the vernalization stage, while the expression of its target gene, ABAR, was downregulated, indicating that temperature-mediated changes in miR395 expression level were associated with vernalization [44]. Among the ten temperature-responsive miRNA families identified in this study, the miR156 and miR167 families were all highly expressed during the nonthermogenic stages, which had lower temperatures than the thermogenic stages. The miR166, miR169, miR171, miR395, and miR399 families had high expression during the thermogenic stages with a higher temperature. The remaining three miRNA families (miR160, miR164, and miR319) had highly expressed members in both the thermogenic stages and nonthermogenic stages. The members of these miRNA families may have experienced functional differentiation during evolution. According to our integrative analysis of thermogenesis-related miRNA and mRNA, downregulated miR156_2 and upregulated miR395a_5, miR481d, and miR319p during thermogenic stages may play a regulatory role in the cellular respiration of *N. nucifera*. Pyrophosphate-fructose 6-phosphate 1-phosphotransferase subunit beta (PFP), cytosolic enolase 3 (ENO3), V-type proton ATPase 16 kDa proteolipid subunit (ATP6V0C), plasma membrane H^+^-ATPase 4 (PMA4), and malate dehydrogenase (MDH) were all proteins encoded by the target genes of miR156_2. Both PFP and ENO3 are enzymes in the glycolysis process. Moreover, PFP is one of the rate-limiting enzymes that catalyze the conversion between fructose-6-phosphate and fructose-1,6-bisphosphate [45]. ATP6V0C and PMA4 may be involved in ATP synthesis coupled to proton transport. Malate dehydrogenase catalyzes the conversion between malic acid and oxaloacetate in the TCA cycle. Similarly, the enzymes ENO3 and PYK2 (pyruvate kinase 2) in glycolysis were identified as the targets for miR395a_5 and miR481d, respectively. ABCG2, associated with ATPase activity, was identified as the target gene for miR395a_5. In addition, the electron carrier RBOHA, which plays a crucial role in hormone signaling, was identified as the target gene for miR319p. The expression of RBOH can be significantly altered under salicylic acid treatment [46]. Previous studies have reported that salicylic acid regulates heat production activity in voodoo lily [47]. Therefore, the *rboh* gene may be associated with heat production. Overall, four temperature-responsive miRNAs, miR156_2, miR395a_5, miR481d, and miR319p, may play a role in floral thermogenesis by regulating the target genes associated with the cellular respiration pathway (Figure 7). Therefore, our study clearly revealed that temperature responsive miRNAs are involved in the regulation of floral thermogenesis in *N. nucifera*.

## 4. Materials and Methods

### 4.1. Plant Materials

A cultivar ‘E Zilian 1′ (also named ‘Mantianxing’) of *N. nucifera* was used for small RNA-seq and transcriptome sequencing, which was provided by the Institute of Vegetable, Wuhan Academy of Agricultural Science, Wuhan, Hubei, China, and obtained the new plant variety right (CNA20130464.0) granted by the Ministry of Agriculture of the People’s Republic of China. Now, the ‘E Zilian 1′ is deposited at Wuhan National Germplasm Repository for Aquatic Vegetables (30°12′ N, 111°20′ E), Wuhan, Hubei, People’s Republic of China, with the deposition number V11A0692. The thermal images were taken with the infrared thermal imager FLIR-SC660 (FLIR SYSTEMS, USA). The temperature of each receptacle was measured with a needle thermocouple and a RDXL4SD digital thermometer (OMEGA, USA). The whole measurement process was carried out from 05:00 to 06:00 in the morning. The development of *N. nucifera* flowers was divided into five periods according to Grant [25]. The receptacles of these five periods were collected, immediately frozen in liquid nitrogen, and stored at −80 °C.

### 4.2. Small RNA and Transcriptome Sequencing

Total RNA was exacted from the receptacles in stages 1, 2, 3, 4, and 5 using pBIOZOL (BIOER, Hangzhou, China) reagent according to the manufacturer’s specifications. RNA purity was detected by using a NanoDrop 8000 spectrophotometer (Thermo Fisher Scientific, Waltham, MA, USA) and RNA quality was detected by Agilent 2100 (Agilent, Santa Clara, CA, USA). Experiment pipeline steps for small RNA sequencing were as follows: the 3′ end adaptor and 5′ end adaptor were ligated to small RNA which had been enriched and purified, and first strand synthesis was undertaken with the Unique molecular identifiers (UMI) labeled Primer. After PCR amplification, PCR products in the range of 100–120 bp were separated using PAGE electrophoresis. Subsequently, quality-qualified libraries were sequenced on BGISEQ-500. The experimental steps of transcriptome sequencing were as follows: rRNA was removed from exacted total RNA using a biotin-labeled specific probe from the Ribo-Zero TM rRNA Removal Kit (Illumina, San Diego, CA, USA). After fragmenting RNA, the first-strand cDNA was synthesized using random primers and reverse transcriptase from the TruSeq^®^ Stranded kit (Illumina, San Diego, CA, USA), and then the double-stranded cDNA was synthesized using DNA polymerase I and RNaseH. The double-stranded cDNA product was subsequently ligated with “A” base and adaptor, which was followed by PCR amplification. Finally, the constructed cDNA library was sequenced on the Illumina HiSeq platform.

### 4.3. Bioinformation Analysis of the Data

Clean data from small RNA sequencing was obtained by filtering out impurities in the raw data, including tags with low quality, 5′ primer contaminants and poly A, tags without insertion and 3′ primer, and tags shorter than 18 nt. To annotate sRNA, Bowtie2 [48] was used to map clean reads to the *N. nucifera* reference genome (http://lotus-db.wbgcas.cn/) and several sRNA database, such as miRBase, Rfam, siRNA, piRNA, and snoRNA. Cmsearch [49] was used to map clean reads to the Rfam database. To ensure that each sRNA corresponds to only one annotation, we followed this priority rule: miRNA > piRNA > snoRNA > Rfam > other sRNA. After annotation, miRA [50] was used to predict novel miRNAs based on the characteristic hairpin structure of the miRNA precursors. UMI [51] species numbers of sequencing reads were used to calculate the expression level of sRNA. DEGseq [52] calculates differential expression based on an MA-plot. The *p*-value of each gene was then corrected by a multiple hypothesis test using Qvalue. A gene was defined as a DEG when reads number fold change ≥2 and Q-value ≤ 0.001. To predict target gene more accurately, the TargetFinder and psRobot software [53,54] were used, and the intersection or union of the target gene predicted by two software packages was chosen as the final result.

For transcriptome sequencing raw data, the short reads comparison tool SOAP [55] mapped reads to the ribosome database, allowing up to five mismatches, and then removed the reads that had been mapped to the ribosome database. Subsequently, clean reads were obtained by screening dirty raw reads, including reads with adapter and low quality, and reads with an N ratio greater than 10%. RSEM [56] was used to calculate the expression level of the transcripts. To standardize the expression of genes, the standardized method used by RSEM is FPKM. To better understand gene function, novel mRNAs and known mRNAs were annotated through databases, including NR, NT, GO, KOG, KEGG, SwissProt, and InterPro. We used Blast [57] or Diamond [58] to make NT, NR, KOG, KEGG, and SwissProt annotations for mRNA, Blast2GO [59] and NR annotation results to make GO annotations, and InterProScan5 [60] to make InterPro annotations.

Time-series clustering analysis was performed using the Mfuzz package [29] to understand temporal expression patterns of the miRNAs and genes. The parameter for the minimum standard deviation, fuzzifier value and c value were set at 0.05, 1.25, and 9, respectively.

### 4.4. qRT-PCR Verification of High Throughput Sequencing Data

#### 4.4.1. qRT-PCR Verification of miRNAs

Total RNA was exacted from the receptacles in stages 1, 2, 3, 4, and 5 using pBIOZOL (BIOER, Hangzhou, China) reagent according to the manufacturer’s specification. For each miRNA, 1 µg of total RNA was reverse-transcribed using miRNA-specific stem–loop primers and the Revert Aid First Strand cDNA Synthesis Kit (Fermentas, life science, Pittsburgh, PA, USA). All stem–loop primers were designed according to Varkonyi-Gasic et al. [61]. The reaction procedure was 16 °C for 30 min, followed by 60 cycles of 30 °C for 30 s, 42 °C for 30 s, and 50 °C for 1 s, and subsequent heating at 70 °C for 5 min to terminate the reaction. Quantitative real-time PCR (qRT-PCR) of miRNA was performed on an ABI StepOnePlus Real-Time PCR System by using SYBR-green fluorescence. Nn_U6 was used as endogenous controls for miRNAs analysis. The reaction procedure was as follows: 95 °C for 30 s, then 40 cycles of 95 °C for 10 s, 56 °C for 30 s, and 72 °C for 15 s. During the amplification process, a melt curve was determined to ensure specific amplification of the products. Finally, the 2^−△△*C*T^ [62] method was used to calculate the expression level of miRNAs in different developmental stages.

#### 4.4.2. qRT-PCR Verification of mRNAs

For determination of mRNA expression, 1 µg of total RNA and OligodT18 primer was used to synthesize cDNA according to the manufacturer’s instructions for the RevertAid First Strand cDNA Synthesis Kit (Fermentas, USA). All primer sequences were listed at Appendix A. Quantitative real-time PCR (qRT-PCR) of target genes were also performed on an ABI StepOnePlus Real-Time PCR System by using SYBR-green fluorescence. NnEF1 were used as endogenous controls for genes analysis. The reaction procedure and calculation method of expression were performed in the same way as described in the qRT-PCR verification of miRNA section. We used GraphPad Prism 7 software for statistical analysis to the qRT-PCR data [63].

### 4.5. Accession Numbers

All the sequencing clean data of this study are available in the NCBI Sequence Read Archive with the accession number PRJNA548651.

## 5. Conclusions

Taken together, the miRNA expression profiles and the gene expression profiles of the receptacle during five development stages were obtained using high-throughput sequencing in the present study. The 29 thermogenesis-related miRNAs and 3024 thermogenesis-related mRNAs were screened based on their differential expression between the thermogenic stages and the nonthermogenic stages. Further functional analysis showed that the cellular respiration pathway and mitochondrial function may play a critical role in floral thermogenesis in *N. nucifera*. For the miRNA family consisting of 29 thermogenesis-related miRNAs, all families except for miR481 are temperature-responsive families. Additionally, the target genes of miR156_2, miR481d, miR395a_5, and miR319p are related to glycolysis, the tricarboxylic acid cycle, and the mitochondrial electron transport chain. miRNAs may participate in floral thermogenesis by regulating genes associated with cellular respiration. These results show that some temperature-responsive miRNAs are involved in the regulation of floral thermogenesis, which had a characteristic feature of the time-series and spatiotemporal expression patterns in *N. nucifera*, and provided comprehensive gene expression profiles and miRNA expression profiles of the receptacle in the development process, which may facilitate precise research into the stable temperature regulation of *N. nucifera*.

## Figures and Tables

**Figure 1 ijms-21-03324-f001:**
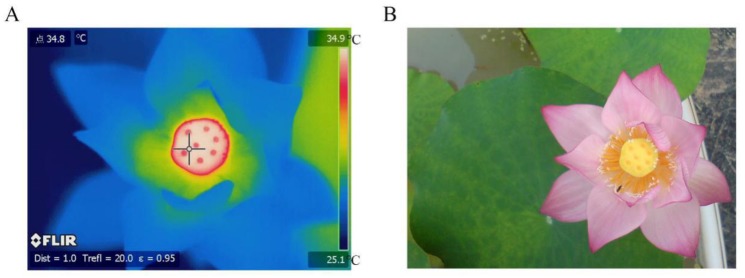
Investigation of floral thermogenesis in *N. nucifera*. (**A**) An infrared image of *N. nucifera* in the thermogenic stage. (**B**) An image of *N. nucifera* in the natural state.

**Figure 2 ijms-21-03324-f002:**
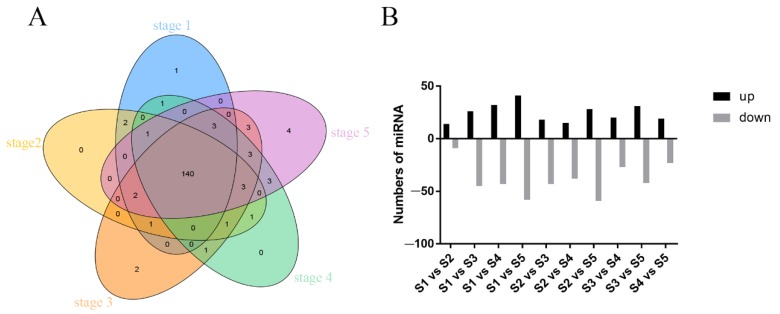
The expression status statistics of miRNAs from five continuous development stages. (**A**) Distribution of the 172 known miRNAs in different development stages of the receptacle. (**B**) Statistics of differentially expressed miRNAs between each two distinct stages. S1, S2, S3, S4, and S5 are the abbreviation of stages 1, 2, 3, 4, and 5, respectively.

**Figure 3 ijms-21-03324-f003:**
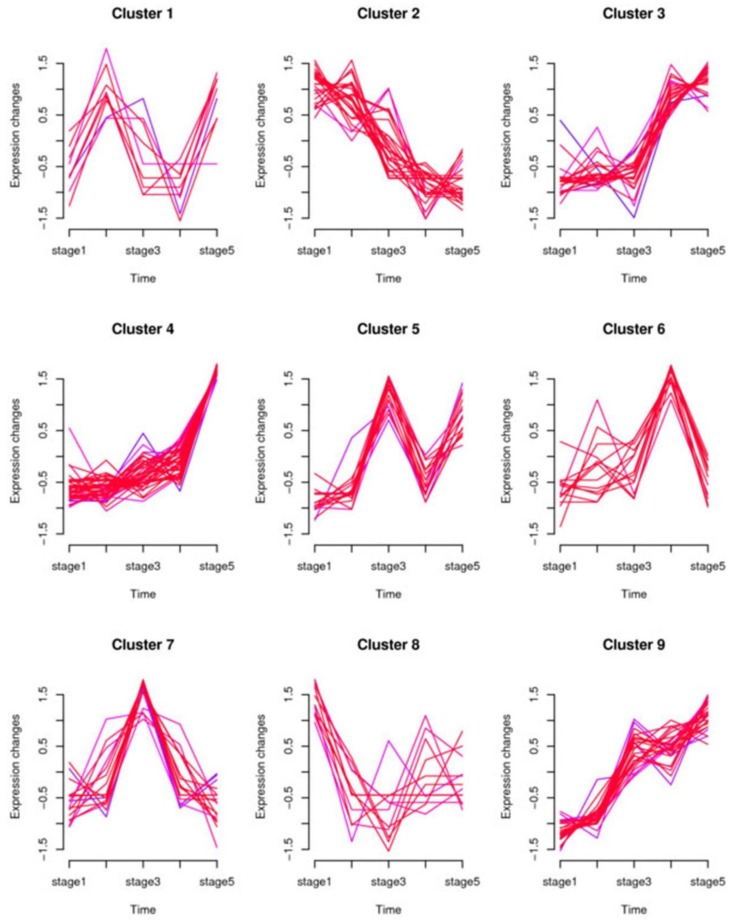
Temporal expression patterns of miRNAs in the five development stages of the receptacle. All miRNAs were classified into nine clusters based on their expression patterns using the Mfuzz R package.

**Figure 4 ijms-21-03324-f004:**
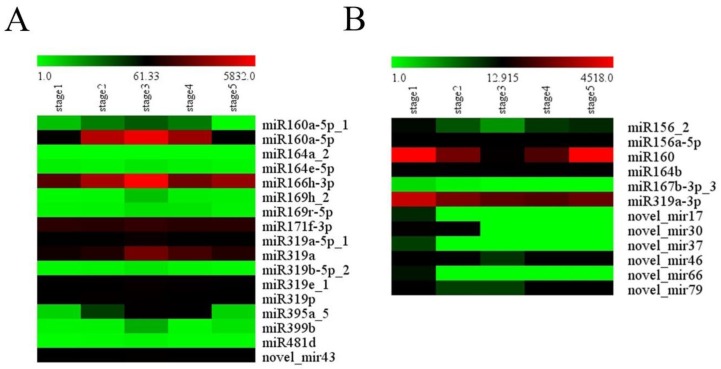
Heat map of miRNA expression in cluster 7 and cluster 8. (**A**) The expression pattern of miRNA in cluster 7. (**B**) The expression pattern of miRNA in cluster 8.

**Figure 5 ijms-21-03324-f005:**
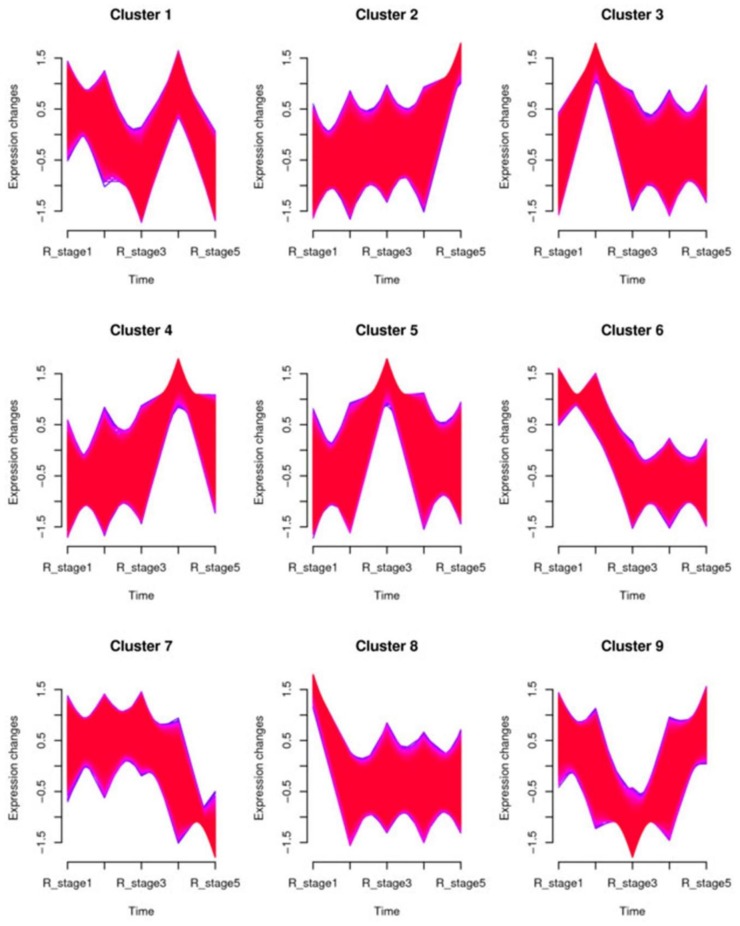
Temporal expression patterns of the differentially expressed mRNAs in five development stages of the receptacle.

**Figure 6 ijms-21-03324-f006:**
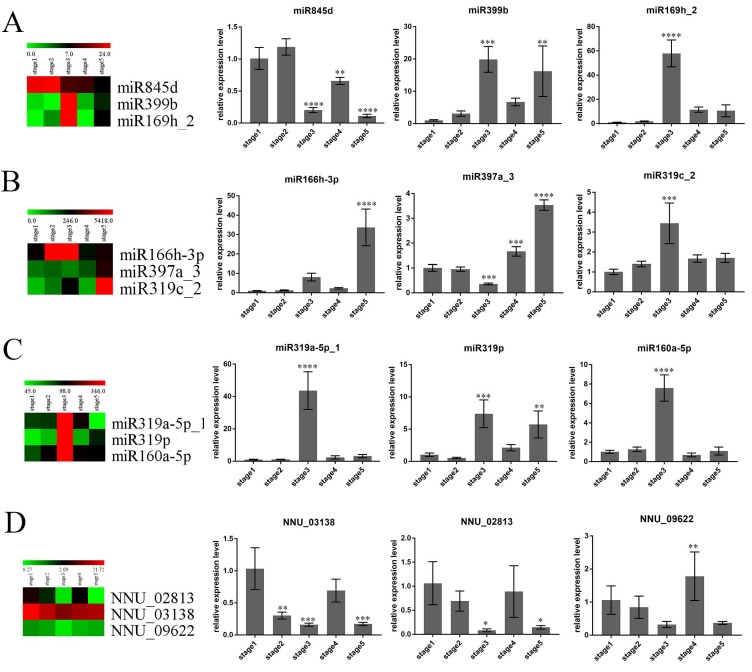
Validation of nine differentially expressed miRNAs and three target genes by qRT-PCR. (**A**) Heatmap and relative expression levels of miR845d, miR399b, and miR169h_2. (**B**) Heatmap and relative expression levels of miR166h_3p, miR397a_3, and miR319c_2. (**C**) Heatmap and relative expression levels of miR319a-5p_1, miR319p, and miR160a-5p. (**D**) Heatmap and relative expression levels of NNU_02813 (the target of miR319p), NNU_03138 (the target of miR319a-5p_1), and NNU_09622 (the target of miR160a-5p). Asterisks indicate a significant difference as determined by one-way ANOVA method between other stages and stage 1. (* *p* < 0.05; ** *p* < 0.01; *** *p* < 0.001; **** *p* < 0.0001.)

**Figure 7 ijms-21-03324-f007:**
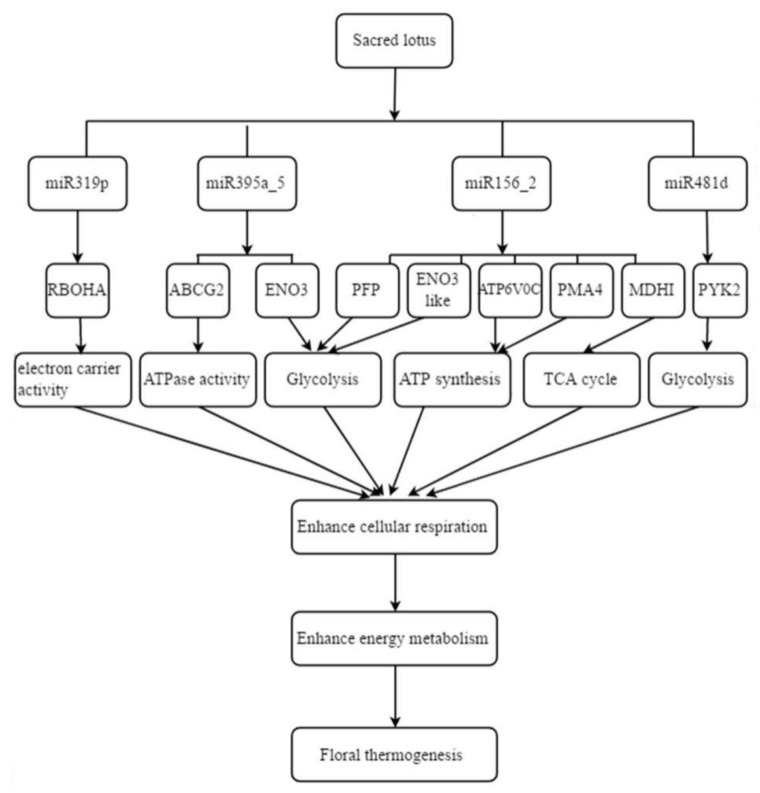
Putative miRNA regulatory network of floral thermogenesis in *N. nucifera*.

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
