# Peer review of "Small RNA and Transcriptome Sequencing Reveals miRNA Regulation of Floral Thermogenesis in Nelumbo nucifera"

_ijms, 2020, doi:10.3390/ijms21093324_

Round 1

Reviewer 1 Report

Please see attached

Author Response

Response to Reviewer 1 Comments

Point 1: The authors need to bring more relevance to the significance of the study in Nelumbo nucifera in both the introduction and the discussion section. 

Response 1: According to your suggestion, we added more content about the significance of the study in Nelumbo nucifera in both the introduction and the discussion section in revised manuscript.

Point 2: This last one needs to highlight also the generality of the finding in relation to the complexity of the phenomenon and the ovary and its function

Response 2: In line 317-323, We emphasized the complex regulatory mechanism of receptacle function and described the effect of floral thermogenesis on pistil function.

Point 3: Figure 4 and 6 are of limited interest due to the generality of the outputs and they are suggested as online resources online.

Response 3: According to your suggestion, we have removed Figure 4 and Figure 6 from the manuscript, and put them in supplementary material 4 and 7,and reordered them as Figure S1 and Figure S3, respectively.

Point 4: Throughout the manuscript please ensure that numbers and units are separated by a space, for instance line 79 and 83 require spacing.

Response 4: We have corrected the error and set a space between numbers and units in line 79 and 83.

Point 5: Throughout the manuscript please ensure that gene names are italicised.

Response 5: Thank you for your suggestion, we have italicized all the gene names.

Point 6: Line 33: “……. reported to be the thermogenesis plants” change to “……. reported to be thermogenic plants”

Line 34 “……….indicated that the thermogenesis….” change to “…… indicated that thermogenesis…”

Line 82 “A great deal of heat” is a colloquialism, change to “A significant/conspicuous amount of heat”

Line 197 ”….. at the five development stages…” it is either “….at the development stage five..” or “…at the fifth developmental stage..”

Line 296: “….such as A. concinnatum, Magnolia denudate and skunk cabbage.” Change to : “…such as A. concinnatum, Magnolia denudata and skunk cabbage.”

Lines 391-392. “In Arabidopsis, six environmental temperature-responsive miRNA families were identified using the microarray experiment and the northern hybridization experiment at 16°C and 23 °C.” Change to “In Arabidopsis, six environmental temperature-responsive miRNA families were identified using microarray and the northern hybridization experiments at 16°C and 23 °C.”

Line 402, the degree symbol (°) should be attached to the C symbol.

Response 6: Thank you for your suggestion. We have corrected the grammatical errors in the above sentence and marked the changes.

Point 7: Figure 3 and 5, axis units and words are rather pixeled, improve the quality/dpi.

Response 7: According to your suggestion, we adopted the TIFF file formats for Figure 3 and 5, and adjusted the resolution of all the images from 300 dpi to 400 dpi.

Point 8: Figure 4 and 6. Limited value should be considered as online resources only.

Response 8: According to your suggestion, we have removed Figure 4 and Figure 6 from the manuscript, put them in supplementary material 4 and 7 and reordered them as Figure S1 and Figure S3 respectively.

Reviewer 2 Report

The authors present an analysis of small RNA and transcriptome during flower development, accompanied by a thermogenesis process, in Nelumbo nucifera.

The thermogenesis process during flower development was studied in other plant species. This article presents the first miRNA analysis in Nelumbo in this process. The results have a preliminary character and can be used as a preface for further research.

The authors undoubtedly put a lot of effort into research. The results have been clearly described and well discussed. However, to improve the manuscript and increase its attractiveness, a few things can be done. Here is the list of my comments and suggestions:

  1. Figure 4 B and D, Figure 6 B and D should be improved. They are of poor quality.
  2. Table 1 and 2 should be moved to the supplementary material.
  3. Instead, heatmap of DEM miRNAs included in cluster 7 and 8 should be shown. This would show readers what miRNAs identified by the authors appear in these clusters.
  4. The authors also identified novel miRNAs. Can any of them be potentially involved in the thermogenesis process?
  5. Validation of transcriptomes by testing the expression of only three genes is definitely not enough. I think it should be just like small RNA-seq.

Author Response

Response to Reviewer 2 Comments

Point 1: Figure 4 B and D, Figure 6 B and D should be improved. They are of poor quality.

Response 1: Based on your suggestions, we adopted the TIFF file formats for Figure 4 and 6, and adjusted the resolution of all the images to 300 dpi. However, according to another reviewer's request, we have put Figure 4 and Figure 6 into supplementary materials 4 and 7 and reordered them as Figure S1 and Figure S3, respectively.

Point 2: Table 1 and 2 should be moved to the supplementary material.

Response 2: Ok, we have deleted Table 1 and Table 2 in revised manuscript, and put them into Supplementary file 2: Table S2 and Supplementary file 5: Table S12, respectively.

Point 3: Instead, heatmap of DEM miRNAs included in cluster 7 and 8 should be shown. This would show readers what miRNAs identified by the authors appear in these clusters.

Response 3: According to your suggestion, the heatmap of DEM miRNAs included in cluster 7 and 8 have been shown in Figure 4.

Point 4: The authors also identified novel miRNAs. Can any of them be potentially

involved in the thermogenesis process?

Response 4: In this study, we performed time-series miRNA expression analysis based on the expression of all identified miRNA. Only miRNAs in clusters 7 and 8, whose expression trends are related to heat generation trend, are considered to be potential regulatory role during floral thermogenesis in Nelumbo nucifera. As shown in Supplementary file 4: Table S6, these two clusters contain eight novel miRNAs, where cluster 7 contains novel_mir43 and novel_mir81, cluster 8 contains novel_mir17, novel_mir30, novel_mir37, novel_mir46, novel_mir66, and novel_mir79. The functions of target genes regulated by these novel miRNAs are shown as follows (Attachment Table1). Among them, novel_mir43, novel_mir81 and novel_mir46 did not predict the target genes.

Attachment Table 1 Thermogenesis-related novel miRNAs and their target genes

miRNA id

Target id

Description

novel_mir17

NNU_04417

beta-aspartyl-peptidase (threonine type)

novel_mir17

NNU_09268

interleukin-1 receptor-associated kinase 4

(ATP binding)

novel_mir17

NNU_20131

methionyl aminopeptidase

novel_mir30

NNU_16530

DNA-dependent ATPase activity

novel_mir30

NNU_24764

rhamnogalacturonan endolyase

novel_mir37

NNU_04664

solute carrier family 32

novel_mir66

NNU_08464

UDP-sugar pyrophosphorylase

novel_mir66

NNU_12236

serine/threonine-protein kinase TNNI3K

(ATP binding)

novel_mir66

NNU_13114

phosphoribosylanthranilate isomerase

novel_mir79

NNU_01188

protochlorophyllide reductase

novel_mir79

NNU_01716

methionyl aminopeptidase

novel_mir79

NNU_04148

SWI/SNF-related matrix-associated actin-dependent regulator of chromatin subfamily A member 2/4

novel_mir79

NNU_05079

respiratory burst oxidase

novel_mir79

NNU_05530

ent-kaurene oxidase

novel_mir79

NNU_07737

Ca2+-transporting ATPase

novel_mir79

NNU_10106

NAD+ kinase

novel_mir79

NNU_13446

calcium-dependent protein kinase

novel_mir79

NNU_20045

glycerol-3-phosphate O-acyltransferase 3/4

novel_mir79

NNU_22346

respiratory burst oxidase

novel_mir79

NNU_23174

DNA repair protein REV1

novel_mir79

NNU_24029

ion channel POLLUX/CASTOR

novel_mir79

NNU_24311

Ca2+-transporting ATPase

novel_mir79

NNU_25334

phosphatidylinositol glycan, class P

As can be seen from the table 1 above, the target genes regulated by novel_mir17, novel_mir30, novel_mir66, and novel_mir79 are all related to ATP synthesis or cellular respiration, which is crucial for heat generation. Therefore, these novel miRNAs may be potentially involved in the thermogenesis process.

Point 5: Validation of transcriptomes by testing the expression of only three genes is definitely not enough. I think it should be just like small RNA-seq.

Response 5: The influencing factors of transcriptome sequencing results mainly include the following aspects:
(1) RNA degradation will seriously affect the quality of sequencing.
(2) Insufficient RNA starting quantity affects the quality of transcriptome sequencing.
(3) The presence of polyA polymer in the library will interfere with the sequencing signal.
(4) Due to the inconsistent gene abundance in the transcriptome, high-abundance expressed genes will obscure low-abundance expressed genes.

In this project, all sequencing samples strictly comply with the detection standards of BGI gene sequencing samples and avoid the occurrence of four situations above to the greatest extent. The results of qRT-PCR for three randomly selected genes are all consistent with the results of sequencing data. These phenomena indicated the high reliability of transcriptome sequencing data to a certain extent. According to your suggestion, validation of transcriptomes by testing the expression of genes should be just like small RNA-seq. However, due to the impact of COVID-19, we are temporarily unable to go back to school for experiments. Therefore, if we have enough time, we could perform verification about the expression of more mRNAs by qRT-PCR based on the high reliability transcriptome sequencing data.